# Bi Objective Peer-to-Peer Ridesharing Model for Balancing Passengers Time and Costs

Seyed Omid Hasanpour Jesri  and Mohsen Akbarpour Shirazi *

Department of Industrial Engineering & Management Systems, Amirkabir University of Technology, Tehran 1591634311, Iran; hasanpoor@aut.ac.ir
* Correspondence: akbarpour@aut.ac.ir

**Abstract:** Ride-sharing services are one of the top growing sustainable transportation trends led by mobility-as-a-service companies. Ridesharing is a system that provides the ability to share vehicles on similar routes for passengers with similar or nearby destinations on short notice, leading to decreased costs for travelers. At the same time, though, it takes longer to get from place to place, increasing travel time. Therefore, a fundamental challenge for mobility service providers should be finding a balance between cost and travel time. This paper develops an integer bi-objective optimization model that integrates vehicle assignment, vehicle routing, and passenger assignment to find a non-dominated solution based on cost and time. The model allows a vehicle to be used multiple times by different passengers. The first objective seeks to minimize the total cost, including the fixed cost, defined as the supply cost per vehicle, and the operating cost, which is a function of the distance traveled. The second objective is to minimize the time it takes passengers to reach their destination. This is measured by how long it takes each vehicle to reach the passenger's point of origin and how long it takes to get to the destination. The proposed model is solved using the AUGMECON method and the NSGA II algorithm. A real case study from Sioux Falls is presented to validate the applicability of the proposed model. This study shows that ridesharing helps passengers save money using mobility services without significant change in travel time.

**Keywords:** ridesharing; vehicle routing; bi-objective optimization; vehicle assignment; AUGMECON; NSGA II; sustainable transportation

## 1. Introduction

The sharing economy is changing our transportation systems. Soon, on-demand services will support flexible mobility and connect people and goods to various transportation systems, including short-distance carpooling and bike or scooter sharing services. Among these services, ridesharing is especially interesting for passengers and public mobility service providers because it addresses issues traditional mobility systems cannot resolve [1]. Ridesharing is a system that allows passengers to share vehicles for similar routes. It can reduce traffic congestion and energy consumption by reducing the number of cars in public spaces and large cities [2]. It can also reduce the time spent searching for parking and travel costs [3]. Additionally, ridesharing has high positive impacts on environmental factors such as greenhouse gas (GHG) emissions reduction [4,5].

Based on the idea of sustainable mobility, bike sharing is great solution for transportation system developments. In [6] the effects of built environment features on bike sharing service are discussed. The findings of this research suggest some recommendations for sustainable development of bike sharing. One of the suggestions is expansion of the bike sharing services in the suburbs or places where proper public transports are no longer reachable. In other research, Elzbieta [7] studied the influencing factors on bike sharing systems. In this research, 25 factors from social, technological, economic, environmental,

and political are recognized and classified based on influences and dependencies. These studies demonstrate bike ridesharing effects on sustainable transportation development.

An increase in the number of vehicles carrying passengers could significantly reduce travel costs for users, such as delays in reaching their destination, and provide time-saving transportation services. However, as the number of vehicles increases, so too do ancillary costs. These are costs directly related to the system's operation; they consist of fixed costs, i.e., supply costs per vehicle, and operating costs, which are a function of vehicle distance traveled. On the other hand, the provision of fewer vehicles results in passengers facing a longer travel time and, consequently, their dissatisfaction. These problems also occur in shared autonomous vehicles in ride-sharing systems, where the main challenges are the system side cost and the passenger side cost [8]. In the system considered in this paper, providers tend to serve all passengers to minimize the total cost for providers and travelers.

Some works such as [9,10] assume that drivers have a fixed and single direction of travel with specific origins and destinations, and in some other cases, detours might occur to pick up and drop off travelers outside the main route, which cannot be a practical assumption for the actual transportation network pattern. To provide more flexibility to ride-sharing services, we assume in this paper that multiple heterogeneous vehicles can operate simultaneously in an area with different travel routes. We also assume that vehicles can pick up passengers with different origins and destinations. In this case, no backward movement from destination to origin points is in any MODM solution allowed for vehicles; however, the vehicles first collect passengers at one or more origin points then take them to one or more destination points in any order. To reduce passenger waiting time, service providers analyze the travel time and set the departure schedule before the vehicle leaves the parking lot [11]. Once a driver accepts a request, the service providers use the travel times and road conditions to find the optimal route to offer the best itinerary to travelers. The cumulative waiting time can be significantly reduced by this method. Against this background, in this work, we investigate optimal ride-sharing routing among different origins and destinations to minimize the total travel time and cost and assign the optimal number of vehicles to passengers. That is, we aim to find a compromise between the optimal number of vehicles and the minimum travel time. As the proposed model is an extension of multi-objective shortest path problem, it can be shown it is a NP-hard problem [12].

The rest of this paper is organized as follows. In Section 2, related works are reviewed. In Section 3, a mathematical model based on MIP is developed. Then, the two multi-objective solution methods, augmented epsilon constraint (AUGMECON) and non-dominated sorting genetic algorithm 2 (NSGA II), are discussed in Section 4. Numerical experiments are performed and evaluated to prove the effectiveness of the proposed model in Section 5. Finally, conclusions are drawn in Section 6.

## 2. Related Work

The mobility-on-demand service has led to ridesharing to address road congestion and facilitate traveling by sharing rides for passengers with similar schedules [9]. Kornhauser [13] was the first researcher who studied ride-sharing systems to help fuel management in the United States and implement a new urban transportation system. This study showed that the use of ride-sharing systems noticeably reduced the energy consumption of taxis, and the work attracted much attention from researchers for different reasons, including reducing the waste of energy [14] relieving traffic congestion [10,15] adjusting prices dynamically [16], and increasing customer satisfaction [17]. Other research regarding ridesharing can be found in [18,19]. Machado recently conducted a comprehensive review of mobility sharing, which considers car-sharing, personal vehicle sharing, bike-sharing, and ridesharing. MENG [19] studied the characteristics of ridesharing and its impact on travel efficiency due to road congestion reduction. This paper also addressed the selected route for a ride-sharing trip, types of ridesharing, travel costs and potential savings, the departure time for a ride-sharing trip from the origin point, and the travel time for a ride-

sharing trip from the origin point. The different types of ride-sharing are also presented in literature which contain ride-sharing with static requests that are determined before the trip begins [20]; ride-sharing with dynamic requests [21], where new requests can be added during the trip; and ride-sharing with both deterministic and stochastic trip requests [22].

However, some challenges are associated with these on-demand services such as travel distance, travel cost, and travel time, which can be handled by implementing optimal vehicles routing. Literature shows these challenges were the case of study in several research studies and authors used different types of mathematic programming to deal with them.

Recent studies show that travel time is a crucial factor in ride-sharing systems, and matching models consider time constraints in finding suitable rides [2]. However, in most of the studied works, a time window was considered in vehicle routing optimization [3,23], or travel time was considered deterministic [24–28]. To minimize the travel cost, [29] presented an optimization model for ride-sharing routes and cost-sharing problems. The problem of maximizing the total number of passengers served was mathematically formulated by [30], and a decomposition algorithm was implemented to solve the model. Some other researchers have tried to minimize the cost function corresponding to the assignment of trip requests to drivers by a combinatorial optimization problem [31]. In [32], a mixed-integer optimization was proposed to optimize ride-sharing routes between certain regions during a particular period. Capacity restrictions were incorporated to ensure that at least each car serves a minimum number of passengers. In this case, they solved the problem by using a genetic algorithm.

Some other researchers used assignment problem to find the optimal assignment of passengers to vehicles, which aimed to address travel cost, time, and distance challenges. The problem of optimal allocation of requests to drivers to minimize the cost function defined as the sum of travel delays over all passengers was solved by [33]. Fielbaum [2] analyzed the positive aspect of optimizing the pick-up and drop-off points in the ride-sharing system. Moreover, he formulated an optimization problem that aimed to minimize the total cost, contained the cost for both drivers and riders, which included the cost of refusing a ride, the additional cost of drivers detouring around passengers' requests, and the operating cost. This model also considered thresholds for maximum waiting time, delay, and walking time (access time). The allocation of vehicles was also considered in the design of the fleet management system [1]. The authors developed a multi-objective integer linear programming (ILP) problem with an algorithm based on the idea of the branch-and-bound algorithm. They also introduced three efficient heuristic methods to speed up the algorithm for large problems. Their considered method aims to optimize all passengers' waiting time and travel time and the total distance and travel time for vehicles. In [34], the authors focused on an optimization model that enables ride-sharing users to walk to and from alternative pick-up and drop-off locations. A late acceptance large-scale and metaheuristic method was presented to solve the model. This problem then presented in order to minimize the total distance traveled. A study focusing on flexible users also showed that walking to and from nearby pick-up and drop-off locations in the ride-sharing system could greatly reduce user rejection [2].

Dynamicity of demands and system condition are studied in some research studies. Adaptive route selection based on the requests of passengers was studied to adjust the path based on passengers' dynamic demands [35]. A learning process method was presented for route selection based on the experiences of the traffic and 'passengers' information [36]. In another study, an agent-based model was presented for the assignment of the vehicles through aggregation of the demand in origins and destinations [37].

The other challenges of the ride-sharing system are related to the number of vehicles in use and fleet allocation and the operational cost regarding these automated vehicles. An efficient ride-sharing service must strike a balance between reducing costs for passengers and the optimal number of vehicles. A large number of vehicles leads to traffic congestion and makes higher costs for providers to serve passengers due to their operating and fixed

costs. To address the challenge of vehicle congestion and determine the optimal distribution of vehicles for a mobility-on-demand system, Wallar [38] developed an algorithm that calculates routes in real-time to ensure that each vehicle is accessible in the available time. A two-stage stochastic optimization model by [35] was provided to decide the allocation of vehicles to each traffic point by maximizing the total profit of the ride-sharing operator, assuming that the departure time of the vehicles is uncertain. In the first phase, an optimization mechanism was used to decide the strategic planning, and in the second phase, an agent-based simulation model tracked the movement of vehicles. A multi-objective optimization problem has been developed to study the operational aspect related to the shared autonomous vehicle (SAV) system using the ride-sharing method [8]. A linear programming problem model was developed in which minimizing the number of SAVs, the sum of travel time, the total travel distance, and the cost of infrastructure construction is assumed to be the objectives of the proposed problem. It should be noted that this paper does not calculate the waiting time for the SAVs to arrive at the passengers' origin as part of the total travel time of the passengers. Moreover, this paper considers the travel time of the SAV rather than the travel time of the passengers. The concept of the autonomous mobility-on-demand system was presented in [39]. In this review, the authors addressed methods and mathematical problems such as dispatching, routing, rebalancing, and ridesharing in relation to autonomous vehicles.

Peer-to-peer ridesharing is another research area that make ridesharing more applicable. Zhoe et al. [40] presented a dynamic decentralized ridesharing platform for vehicle assignment and routing by utilizing a traffic prediction module. They also formulated a mathematical model to assign the vehicles to requests and corresponding routes for candidate vehicles and paths. However, in the proposed model each vehicle could not serve more than one request in any time interval, i.e., two passengers with similar origins and destinations could not be picked up by the same vehicle in a common time interval. Ramon et al. [41] proposed a framework for designing on demand multimodal transit system which integrated ridesharing in network design. It formulated two optimization models for network designs and fleet sizing optimization. The first model determined bus lines and routes of shuttles and the second model optimized the number of shuttles; meanwhile, the set of shuttle routes as ridesharing vehicles were given as an input of optimization model and the model solved the route assignment problem instead of vehicle routing. In [42], a dynamic tree algorithm for solving ride-sharing problem was introduced to match peer-to-peer demands. They have defined a local accessible region for any driver and the passengers who have the pickup and drop-off locations inside the accessible region could be considered as potential requests for ridesharing. So, the assignment of passengers to a vehicle is based on a local search, while in our proposed method a holistically and integrated passenger assignment and vehicle routing problem are applied. In [34], for each driver a set of feasible routes were defined in which passengers were picked up and dropped off and the origin and destinations of drivers were fixed and detours were assigned to picked up and drop off the passengers. The summary of most related researches is presented in Table 1.

To the best of our knowledge, most researchers used decentralized models for making decision about fleet allocation, vehicle routing, and the passenger's assignment. In this research, we have presented an integrated routing, fleet allocation, and the passenger's assignment in a centralized optimization model that optimized all holistically. Analyzing the effect of the number of assigned vehicles to the passenger's cost and time as the most important criteria is another contribution of this research, while it offers a set of non-dominated solutions to passengers for decision making based on passengers' preferences. Moreover, our model can be extended to other types of mobility-on-demand systems, such as shared autonomous vehicle systems. Likewise, the capacity and operating costs of different vehicles could vary so that it could be applied to transportation systems with multiple types of vehicles.

**Table 1.** Summary of the literature.

| Literature | Investigated Problem | Proposed Method | Objectives |
| --- | --- | --- | --- |
| (Fielbaum, 2021) [2] | Assignment problem | Heuristics | Minimizing passenger cost, penalty, extra passenger cost, and the operational cost. |
| (Alonso-Mora, 2017) [33] | Assignment problem | reactive anytime optimal method | Minimizing travel delay cost |
| (Alisoltani, 2021) [1] | Assignment problem | Heuristic Branch and bound | Minimizing distance for vehicles and waiting time and total travel time for passengers |
| (Seo, 2021) [8] | Autonomous vehicle (SAV) system design problem | weighted sum method | Minimizing total travel time, total travel distance, total numbers of SAVs, and infrastructure construction cost |
| (Smet, 2021) [34] | Generalized Vehicle Routing Problem | The late acceptance and meta-heuristic method | - |
| (Long, 2018) [22] | Stochastic ride-sharing problem | Monte Carlo simulation (MCS) | Maximizing total generalized trip cost-saving and the number of matches |
| (Huang, Kai and Huang, Yantao and Kockelman, Kara M, 2022) [35] | vehicle assignment problem under departure time uncertainties | two-phase stochastic optimization model | Maximizing the total profits of vehicle sharing operator |
| (Cao, 2021) [32] | Ride-sharing route problem | Genetic and branch and bound algorithm | Minimizing total travel distance |
| (Bei, 2018) [31] | Assignment problem | approximation algorithm | Minimizing cost function |
| (Masoud, 2017) [30] | peer-to-peer multi-hop ride-matching problem | decomposition algorithm | Maximizing total number of served passengers |

## 3. Mathematical Formulation

A transportation network is considered as a directed graph $G(N, A)$, where the number of nodes is $n = |N|$, and it is divided into depot nodes ($O$), passenger nodes ($N^P$), and destination nodes ($N^D$). The set of vehicles $V = \{1, ..., v, ..., V\}$ represents the maximum available vehicle. It is clear that the optimal number of vehicles is one of the model's outputs. The passenger set includes the number of passengers ($p$), the origin of passengers ($n^P$), and the destination of passengers ($n^D$). The decision variable $S_p^v$ would be 1 if passenger $p$ is picked up by vehicle $v$ and $x_{ij}^k$ is 1 if vehicle $v$ travels from node i to node j. The variable $t_i^v$ measures the arrival time of vehicle $v$ at node $i$. The capacity and fixed cost of the vehicle ($i$) are represented by $Q(v)$ and $FC(v)$. It is assumed that the arrival time of all passengers at their origin points is known and that passengers arrive at their origin points before the vehicles are dispatched. All sets, parameters, and decision variables of the model are listed in Table 2.

**Table 2.** The set, parameters, and decision variables of the model.

| Set and Parameters | |
| --- | --- |
| $P, p$ | The set and index of all passengers |
| $V, v$ | The set and index of all vehicles |
| $origin(p, n^P)$ | The origin matrix with values 0 and 1; If passenger p is in node $n^P$, the corresponding value takes the value 1, otherwise it takes the value 0. |
| $destination(p, n^D)$ | The destination matrix with values 0 and 1; If $n^D$ is the destination of passenger $p$, the corresponding value takes the value 0. |
| $N^P$ | The set of passengers' locations |
| $N^D$ | The set of all destinations |
| $o$ | The Depot node |
| $N$ | The set of all nodes in the network (including passengers' locations, passengers' locations, and Depot node, $N = N^P \cup N^D \cup o$) |

| | |
|---|---|
| $c_{ij}$ | The traveling cost between two nodes $i$ and $j$ ($i, j \in N$) |
| $T_{ij}$ | The traveling time between two nodes $i$ and $j$ ($i, j \in N$) |
| $FC(v)$ | The fixed shipping cost of vehicle $v$ |
| $Q(v)$ | The capacity of vehicle $v$ |
| $T_p$ | The time of arriving passenger $p$ to its origin |
| **Decision variables:** | |
| $x_{ij}^k$ | 1, if vehicle $k$ goes from node $i$ to node $j$, 0 otherwise ($i, j \in N$). |
| $S_p^v$ | 1, if passenger $p$ is picked up by vehicle $v$, 0 otherwise ($p \in P$). |
| $t_i^v$ | The time when vehicle $v$ reaches to node $i$. |

The mathematical formulation for our model is as follows:

$$Z_1 = Min \left\{ \sum_v \sum_{j \in N} FC(v)\, x_{o,j}^v + \sum_v \sum_{i \in N} \sum_{i \in N} C_{ij} x_{i,j}^v \right\} \tag{1}$$

$$Z_2 = Min \sum_{d \in N^D} \sum_{p \in P} S_p^v \times destination\left(p,\, n^D\right) \times \left(t_d^v - T_p\right) \tag{2}$$

Subject to:

$$\sum_{v \in V} S_p^v = 1 \forall\, p \in P \tag{3}$$

$$\sum_{p \in P} S_p^v \leq Q(v) \forall\, v \in V \tag{4}$$

$$\sum_{\substack{i \in N \\ i \neq j}} x_{i,j}^v \geq S_p^v \forall\, j \in N^P,\, \forall\, p \in P,\, \forall\, v \in V : origin(p, j) = 1 \tag{5}$$

$$\sum_{\substack{j \in N \\ j \neq o}} x_{o,j}^v \geq S_p^v \forall\, p \in P,\, \forall\, v \in V \tag{6}$$

$$\sum_{\substack{i \in N \\ i \neq j}} x_{i,j}^v \geq S_p^v \forall\, p \in P,\, \forall\, v \in V,\, \forall j \in N^D : destination(p, j) = 1 \tag{7}$$

$$\sum_{\substack{i \in N \\ i \neq j}} x_{i,j}^v - \sum_i x_{j,i}^v = 0 \forall\, j \in N^P,\, \forall\, v \in V \tag{8}$$

$$\sum_{\substack{j \in N \\ j \neq o}} x_{o,j}^v \leq 1 \forall\, v \in V \tag{9}$$

$$\sum_{i \in N^P} \sum_{j \in N^D} x_{i,j}^v \leq 1 \forall v \in V \tag{10}$$

$$\sum_{\substack{j \in N^D \\ j \neq i}} x_{i,j}^v \leq 1\, i \in N^D, \forall\, v \in V \tag{11}$$

$$t_j^v = \sum_{i \neq j} x_{i,j}^v * \left[t_i^v + T_{ij}\right] \forall\, j \in N, j \neq o, \forall\, v \in V \tag{12}$$

$$t_o^v = 0 \forall\, v \in V \tag{13}$$

$$x_{i,j}^v \leq \sum_{p \in P} S_p^v \forall\, i,j \in N, \forall\, v \in V \tag{14}$$

$$\sum_{j \in N^p \cup o} x_{i,j}^v = 0 \forall i \in N^D, \forall\, v \in V \tag{15}$$

$$\sum_{\substack{i \in N^D \\ i \neq j}} x_{i,j}^v \geq \sum_{\substack{u \in N^D \\ u \neq j}} x_{j,u}^v \forall j \in N^D, \forall\, v \in V \tag{16}$$

$$x_{i,j}^v,\; S_p^v \in \{0,1\},\; t_j^v \in Int+ \forall\, i,j \in N, \forall\, v \in V, \forall\, p \in P \tag{17}$$

As can be seen, the proposed mathematical model contains two objective functions. The first objective function ($Z_1$) minimizes the total cost, including the fixed vehicle and travel costs. The second objective function ($Z_2$) also minimizes the total time required for passengers to reach their destination, including the passengers' waiting time until they are picked up by the vehicles and the passengers' travel time.

Constraint (3) guarantees that each passenger is picked up by exactly one vehicle. Inequality (4) states that the number of passengers in each vehicle must not exceed the vehicle's capacity. Constraint (5) shows that if passenger $p$ is picked up by vehicle $v$, the vehicle must enter passenger $p$'s node. According to constraint (6), if a passenger is picked up by vehicle $v$, the vehicle must be dispatched from origin $o$. Constraint (7) guarantees that if passenger $p$ is picked up by vehicle $v$, the vehicle must enter passenger $p$'s destination node. Equation (8) is a degree constraint where if vehicle $v$ travels to node $i$ $(i \in N^D)$, it should pass through that node. Constraint (9) shows that each vehicle travels at most one route from the origin node. Moreover, according to this constraint, not all vehicles need to be dispatched. Constraint (10) states that each vehicle cannot travel to more than one destination point from all passenger nodes. Constraint (11) ensures that each vehicle can only travel to a maximum of one of the destination nodes and one other destination point from each destination point. Constraint (12) specifies when vehicle $v$ reaches node $j$. This constraint prevents the formation of sub-tours. Based on equation (13), the initial time when leaving the source node is zero for all vehicles. Constraint (14) states that a trip from node $i$ to node $j$ by vehicle $v$ is possible only if vehicle $v$ is sent from the origin. Constraint (15) prevents vehicle $v$ from returning from the destination nodes to the passenger and origin nodes. Constraint (16) states that vehicle $v$ can only leave destination node $j$ if vehicle $v$ has arrived at that destination node. Finally, constraint (17) shows the domain of each variable.

As can be seen, a part of the $Z_2$ is obtained by multiplying a binary variable ($S_p^v$) by an integer variable ($t_d^v$), which is nonlinear. To linearize $Z_2$, we name the multiples of $S_p^v$ and $t_d^v$ as $h_d^v$, and add the following constraints (constraints (18–20)).

$$h_d^v \leq M \times S_p^v \forall d \in N^D, \forall p \in P, \forall\, v \in V \tag{18}$$

$$h_d^v \leq t_d^v \forall d \in N^D, \forall\, v \in V \tag{19}$$

$$t_d^v - h_d^v + M \times S_p^v \leq M \forall d \in N^D, , \forall p \in P, \forall\, v \in V \tag{20}$$

where $M$ is a very large number.

Therefore, according to the above explanations, $Z_2$ changes to the constraint (21).

$$Z_2 = Min \sum_{d \in N^D} \sum_{p \in P} h_d^v \times destination\left(p, n^D\right) - S_p^v \times T_p \tag{21}$$

In constraint (12), which is nonlinear, the multiples of the two variables $x_{i,j}^v$ and $t_i^v$ is called $g_{i,j}^v$, and similar to Bab, we convert it to its linear equivalent based on the following constraints.

$$g_{i,j}^v \leq M \times x_{i,j}^v \forall\, i,j \in N, \forall\, v \in V \tag{22}$$

$$g_{i,j}^v \leq t_i^v \forall\, i,j \in N, \forall\, v \in V \tag{23}$$

$$t_i^v - g_{i,j}^v + M \times x_{i,j}^v \leq M \forall\, i,j \in N, \forall\, v \in V \tag{24}$$

Thus, constraint (12) changes to constraint (25).

$$t_j^v = \sum_{i \neq j} g_{i,j}^v + \sum_{i \neq j} x_{i,j}^v \times T_{ij} \forall\, j \in N, j \neq o, \forall\, v \in V \tag{25}$$

## 4. Multi-Objective Method

Several methods for solving multi-objective optimization problems have been presented, such as goal programming (GP), weighted sum method (WSM), epsilon constraint (EC), augmented epsilon constraint (AUGMECON), and lexicography (Lex) [43]. In addition, meta-heuristic methods (e.g., the non-dominated sorting genetic algorithm (NSGA II)) have been developed to solve complex MODM or large-scale problems [44].

In any MODM solution, whether exact or meta-heuristic methods, the goal is to find an efficient solution set where the values of the objective function are non-dominated and located on the Pareto solution [45]. AUGMECON is an efficient exact method that prevents from weakly pareto solutions and has higher calculation speed rather than epsilon constraint (EC) as it steers clear of redundant iterations. Whereas, multi-objective evolutionary algorithms (MOEA) can speedily discover pareto solutions [46]. Around MOEA algorithm, NSGA II is recognized as an effective algorithm and recommended in the literature [47]. As a result, the AUGMECON method is used as an exact method to obtain exact Pareto solutions and to show the relationship between the objectives and the NSGA II algorithm developed for large-scale problems.

a.    Augmented epsilon constraint method

As mentioned earlier, several methods have been proposed for solving multi-objective problems, with the epsilon constraint (EC) method being one of the most popular. In this method, one of the objective functions is considered the principal function, and the others are applied to the problem as constraints.

Several developments have been presented for EC to make it more efficient. The augmented epsilon constraint (AUGMECON) presented by George Mavrotas is one of the most efficient techniques [46]. AUGMECON includes the following steps:

1st Step:

One of the objective functions is considered principal. Here, the first objective function $(Z_1)$ is principal.

2nd Step:

The problem is solved by considering an objective function, and the optimal value of each objective function is determined. To be specific, the problem with $Z_1$ and $Z_2$ are solved independently.

3rd Step:

The lexicographic method is used to determine second objective function's best and worst solution. Accordingly, the best solution for the second objective function is its optimal value when solved individually as an objective function. Then, the worst value of the second objective function is determined by optimizing it and considering the first objective function as a constraint in its optimal value. Thus, the interval for second objective function is determined.

$$\left[ Z_2^{max}, Z_2^{min} \right] \tag{26}$$

$$r_{2=} Z_2^{max} - Z_2^{min} \tag{27}$$

4th Step:

The interval between two optimal values of the second objective function is divided into a defined $(q_i)$, and a table for the epsilon values is created.

$$\varepsilon_2^k = Z_2^{max} - \frac{r_2}{q_i} * k \; k = 0, 1, \ldots, q_i \tag{28}$$

5th Step:

The problem is solved with the principal objective function by considering each epsilon. Accordingly, the constraints of the secondary objective functions are transformed into equivalent constraints by using shortage or surplus variables. Moreover, the problem is solved, and efficient solutions are generated.

The new problem is defined as follows:

$$min\{ Z_1(x) + \delta * (s_2)\}Z_2(x) = \varepsilon_2 + s_2 x \in X,\ s_i \in R^+ \tag{29}$$

6th Step:

Finally, in each round by solving the new problem and considering different epsilon values, a part of the Pareto solutions is extracted.

b.   NSGA II

One of the most popular and efficient MOEAs is the NSGAII algorithm. It has been presented by Deb et al. [48]. The Pareto solutions are ranked and sorted by non-dominated crowding and sorting distance operators in this algorithm. A double-point crossover and the insertion, swap, and reversion operators for mutation and roulette wheel selection are used to generate offspring from the parent population. Then, each of the objective functions is evaluated individually. Pareto fronts are created by ranking the population based on the non-dominated sorting process. Ultimately, different non-dominated front solutions engross the new population in terms of their ranks. After the completion of each round, a non-dominated Pareto set is captured. Here, we present the pseudocode of the main phases of the NSGA II algorithm in Algorithm 1 [47].

---

**Algorithm 1.** NSGA-II Algorithm.

---

Determining parameters (population size, mutation rate, iteration count)

2:    Generating P random populations
3:    Checking and modifying the feasibility of each individual
4:    Calculating the objectives individually
5:    Calculating the crowding distance based on determining the rank of each solution
6:    chromosome selection by the binary tournament selection
7:    Using mutation operators and crossover
8:    Creating Q offspring
9:       For i = 1 to "iteration count"
10:       for any member of the population
11:       Determining the solution rank
12:       Applying Crowding distance operator in order to sort the last pareto front solutions
13:       end
14:    best solution selection
15:    Creating the upcoming generation
16:    Selection by roulette wheel
17:    Recombining and mutation
18:    end
19:   End

---

c.   Solution representation in NSGA II

Primarily, the assigned passenger to each vehicle must be determined. For this purpose, the first chromosome structure should be designed. A vector is generated from random numbers between 0 and 1, while the size of the vector is equal to the number of passengers plus the number of vehicles minus 1. Then, this vector converts to a vector of integer numbers by arranging and determining the position of each vector member. Moreover, the elements of the new vector that have values less than or equal to the sum of passengers are considered as passenger's position and the other elements (with values greater than sum of passengers) are vehicle positions. The position of vehicles is considered as a separator and

each passenger is assigned to its first right vehicle. For example, if there are four customers and four vehicles, a random vector could look like Figure 1:

| 0.276594 | 0.701511 | 0.580986 | 0.981192 | 0.324432 | 0.954345 | 0.045005 |
|---|---|---|---|---|---|---|

By arranging the above vector:

| 0.045005 | 0.276594 | 0.324432 | 0.580986 | 0.701511 | 0.954345 | 0.981192 |
|---|---|---|---|---|---|---|

By setting the position of each member in the arranged vector:

| 7 | 1 | 5 | 3 | 2 | 6 | 4 |
|---|---|---|---|---|---|---|

| Vehicle position |
|---|
| Passenger positions |

**Figure 1.** The structure of Chromosome.

Therefore, elements 1, 3, and 6 (equal to 5, 6, and 7) are considered as the vehicle positions and separators. According to the designed chromosome, the arrangement and passengers assigned to each vehicle are as follows:

Vehicle 1: No passenger assigned (because there is no passenger on the left of this vehicle)

Vehicle 2: Passenger No. 1

Vehicle 3: Passengers No. 2 and 3

Vehicle 4: Passenger No. 4

The assignment and arrangement of passengers visited in each vehicle are obtained based on the order of elements in the chromosome. Accordingly, the motion path of each vehicle is determined. For this purpose, each vehicle initiates the movement from the depot and travels to the starting point of the passengers in the order of determined arrangement according to the chromosomes. The K-nearest neighbor (KNN) algorithm is used to find the paths of the vehicles to reach the nearest node. The algorithm finds the shortest path between two nodes to get from one passenger origin to the next passenger origin. Furthermore, this algorithm preferably determines the closest passenger destination to the last node visited by the customer, and the vehicle travels to the closest destination. Furthermore, this procedure continues until, first, the origin and then, the destination of all customers associated with the respective vehicle have been visited.

Solution Feasibility

To confirm the feasibility of the chromosome solutions, it is necessary to verify that the number of passengers assigned to each vehicle does not exceed its capacity. For this purpose, we use a penalty function as follows.

$$PF_v = \max\left(0, \frac{UC_v}{Q(v)} - 1\right) \tag{30}$$

In which $UC_v$ is the number of passengers assigned to vehicle $v$, $Q(v)$ is the capacity of vehicle $v$, and $PF_v$ is the amount of capacity violation for vehicle $v$. According to this equation, the amount of violation is calculated when the number of passengers assigned to the vehicle exceeds its capacity. The average violation of all vehicles is multiplied by a large number and added to the objective functions. Therefore, the infeasible solutions are excluded from the algorithm calculation process.

Considering the violation functions, the objective functions are as follows:

$$Z_1 = Z_1' + M\left(\overline{PF}\right) \tag{31}$$

$$Z_2 = Z_2' + M\left(\overline{PF}\right) \tag{32}$$

where $Z_k'$ is the value of the kth objective function, $M$ is the large number, and $\overline{PF}$ is the mean value of the penalty functions.

Mutation

Insertion, swap, and reversion operators are employed for mutation.

Swap

This operator selects two points from the chromosome randomly and substitutes them. An example is presented in Figure 2. The blue and yellow points show the substituent points.

| 0.619 | 0.332 | 0.982 | 0.248 | 0.081 | 0.43 |
|---|---|---|---|---|---|

| 0.619 | 0.081 | 0.982 | 0.248 | 0.332 | 0.43 |
|---|---|---|---|---|---|

**Figure 2.** Swap Operator.

Insertion

This operator randomly selects two points from the chromosome and moves the first point to the right of the second point. An example is presented in Figure 3. The blue point is a random point which moves to the next of second random point (yellow).

| 0.619 | 0.332 | 0.982 | 0.248 | 0.081 | 0.43 |
|---|---|---|---|---|---|

| 0.619 | 0.982 | 0.248 | 0.081 | 0.332 | 0.43 |
|---|---|---|---|---|---|

**Figure 3.** Insertion Operator.

Reversion

This operator randomly selects two points from the chromosome, substitutes them, and reverses the values of the two points. Figure 4 shows an example for reversion operator. The blue and yellow points show the substituent points.

| 0.619 | 0.332 | 0.982 | 0.248 | 0.081 | 0.43 |
|---|---|---|---|---|---|

| 0.619 | 0.081 | 0.248 | 0.982 | 0.332 | 0.43 |
|---|---|---|---|---|---|

**Figure 4.** Reversion Operator.

Crossover

A double-point crossover operator is used for the crossover. In this method, two points are selected, the points between these two points in two parents are shifted, and the children's chromosomes are obtained. In Figure 5 an example of parents and the results of crossover (children) is presented. The red points show the crossover points and the blue and green points are swapping points.

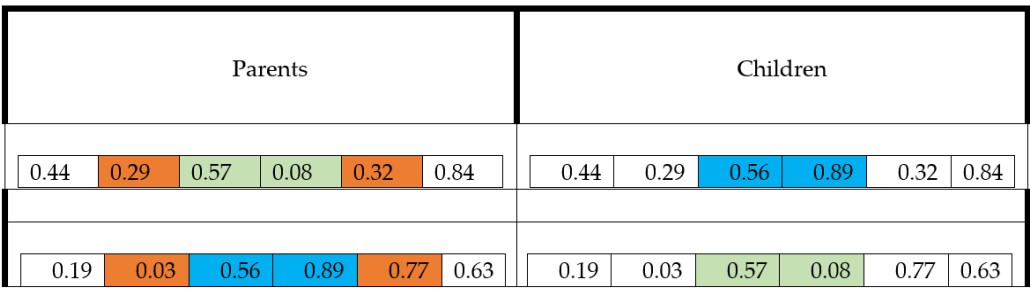

**Figure 5.** Crossover Operator.

## 5. Computational Results

This section presents (1) the small-scale numerical examples and (2) the practical application. In the first section, the results of the AUGMECON methods and the NSGA II algorithm are compared for small to medium cases to measure the quality of the NSGA II solutions. In the second section, a real world case study is solved by NSGA II with different parameters, and the results are analyzed.

a.      Comparison between AUGMECON and NSGA II

The various small- and medium-scale problems are solved by AUGMECON and NSGA II. The AUGMECON method is developed in GAMS, and NSGA II is coded in MATLAB. As mentioned earlier, the problem complexity presented in NP-hard and exact methods such as AUGMECON cannot solve large-scale problems in a reasonable time. An efficient method of solving large-scale problems is a heuristic algorithm such as NSGA II and MOPSO. To test the efficiency of the proposed NSGA II algorithm, different problems with different parameters and size are solved using both methods, the results are compared, and running times are noted. In all problems, the fixed cost for vehicle assignment is assumed to be 4 and 1000, respectively. To understand the results, we start by setting the arrival time of all passengers at the starting points to zero.

To calculate the error between the AUGMECON and NSGA II algorithms, the mean absolute percentage error (MAPE) is calculated. For problems (1) to (7), the results of the two algorithms are exactly the same, and the Pareto solution points are completely compatible. For problems (2), (4), and (7), the NSGA II algorithm found one more Pareto solution. The problem parameters and the comparison between the results of the two algorithms are summarized in Table 3. All tests were performed on a laptop with 12 GB of memory and an i5-1135G7 CPU 2.40 GHz.

**Table 3.** Comparison between AUGMECON and NSGA II.

| Problem Number | Method | Maximum Number of Vehicle | Passenger Number | Network Size | Run Time (s) | Number of Pareto Front Solutions | MAPE (%) |
|---|---|---|---|---|---|---|---|
| 1 | NSGA II | 4 | 3 | 7 | 0.78456 | 3 | 0 |
|   | AUGMECON | 4 |   |   | 8.677 | 3 |   |
| 2 | NSGA II | 4 | 4 | 7 | 0.7451 | 4 | 0 |
|   | AUGMECON | 4 |   |   | 12.241 | 3 |   |
| 3 | NSGA II | 4 | 6 | 7 | 0.87662 | 3 | 0 |
|   | AUGMECON | 4 |   |   | 198.141 | 3 |   |
| 4 | NSGA II | 4 | 8 | 7 | 0.6805 | 4 | 0 |
|   | AUGMECON | 4 |   |   | 388.575 | 3 |   |
| 5 | NSGA II | 4 | 3 | 10 | 0.7764 | 2 | 0 |
|   | AUGMECON | 4 |   |   | 4.79 | 2 |   |
| 6 | NSGA II | 4 | 6 | 10 | 0.7325 | 4 | 0 |
|   | AUGMECON | 4 |   |   | 2866.67 | 4 |   |
| 7 | NSGA II | 4 | 8 | 10 | 1.085 | 3 | 0 |
|   | AUGMECON | 4 |   |   | 17902.4 | 3 |   |
| 8 | NSGA II | 6 | 6 | 10 | 4.37 | 6 | % 0.07 |
|   | AUGMECON | 6 |   |   | 81,063.8 | 5 |   |

The Pareto fronts for the different problems are shown in Figure 6. The red points show the pareto front set of the AUGMECON method and the yellow points are the NSGA II pareto set. The points which are common in both algorithms are green.

The size of problem is growing from problem (1) to (8). In most of the problems, the NSGA II and AUGMECON generated very close pareto front sets. AUGMECON solutions are optimal, but it needs much time for medium or large-scale problem, while NSGA II is capable of finding the optimal or near to optimal solutions in much less than time.

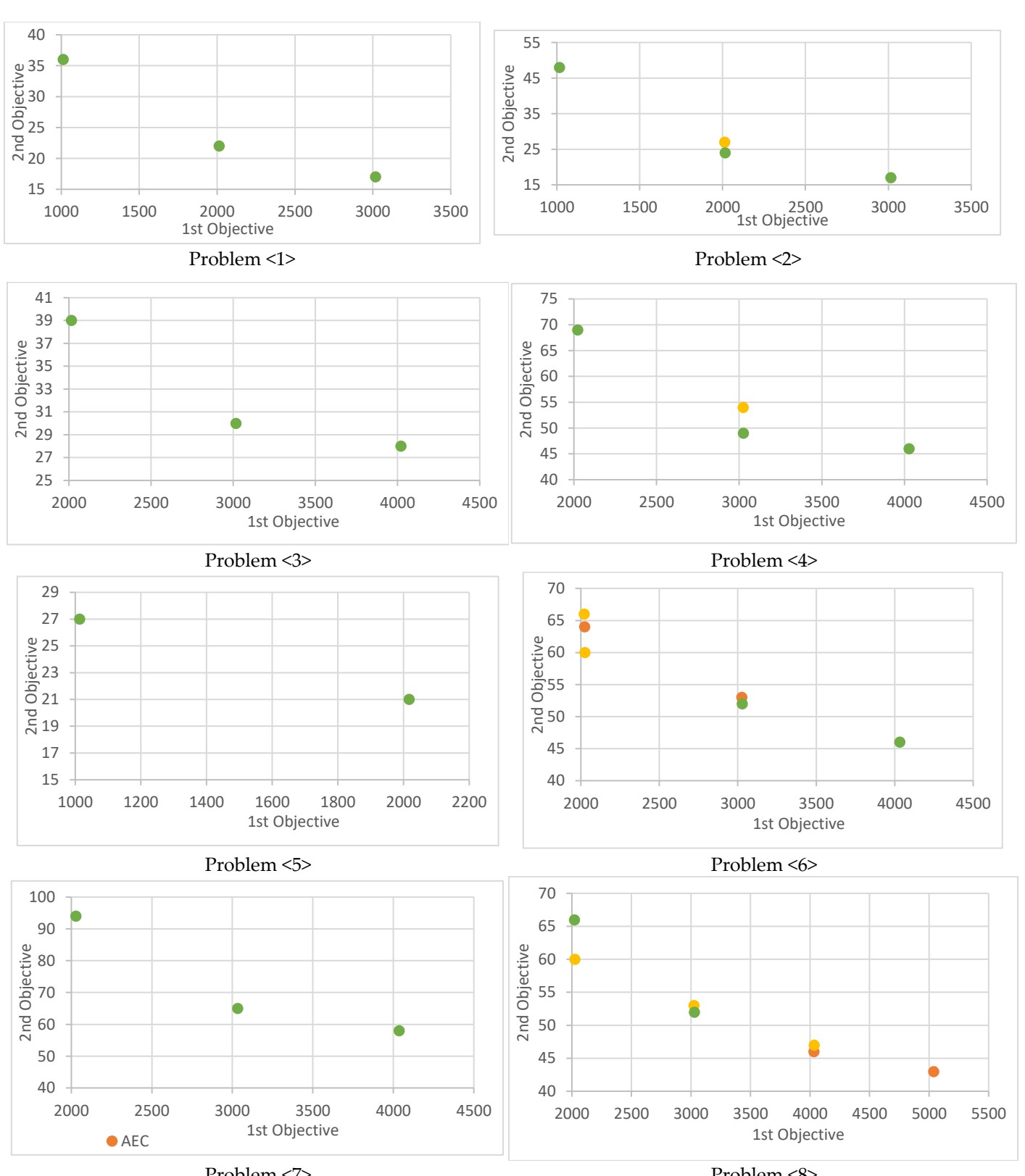

**Figure 6.** The Pareto front for test problems.

To evaluate this model for a problem with 10 nodes, 6 passengers, and 6 available vehicles, we ran it in GAMS for 23 h. However, the method found the optimal solutions, but it could not be an expectable time for real cases which have significantly larger scales. Figure 7 shows the runtime for a problem with 10 nodes, 4 available vehicles, and a

different number of passengers. The larger number of passengers makes the runtime increase exponentially. The runtime of the problem also depends strongly on the number of available vehicles and the size of the network.

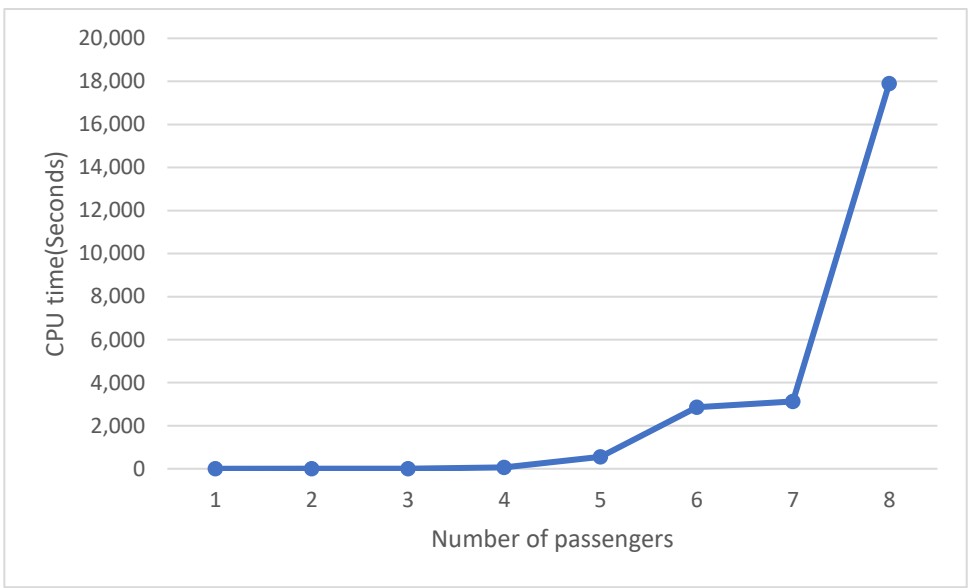

**Figure 7.** Runtime for AECAUGMECON method.

As Table 3 shows, the NSGA II algorithm can find more Pareto-optimal solutions than the AUGMECON method. In addition, it is as accurate as AUGMECON, with much faster simulations run time. For example, the results from a test problem are shown in Section 5 and Figures 9–13. In this problem, node 1 is the depot, node 8, node 9, and node 10 are destinations, and the numbers on the arcs show travel times. It is assumed that travel time is the same as travel cost. Each figure shows the routes of each vehicle and the passengers assigned to that vehicle. For example, in solution 1 (found by both AUGMECON and NSGA II), the first vehicle route is [1-4-5-8], and passenger 5 in node 4 is picked up and delivered in node 8. To calculate the first objective function, we need to first calculate the fixed cost. The fixed cost equals the number of assigned vehicles multiplied by their respective costs: 5 vehicles × 1000 credits. The second cost we need to calculate is the variable cost, which is equal to the sum of the distances traveled by each vehicle, which is 29. The second objective function is the total time of arrival of all passengers. The time taken by the first passenger picked up by vehicle 5 consists of two parts: the waiting time for the vehicle to arrive at node 2, which is equal to 1, and the travel time to arrive from node 2 (origin) to node 8 (destination), which is equal to 3. So, the total time for passenger 1 is equal to 4. The total time for all passengers in solution 1 is 43.

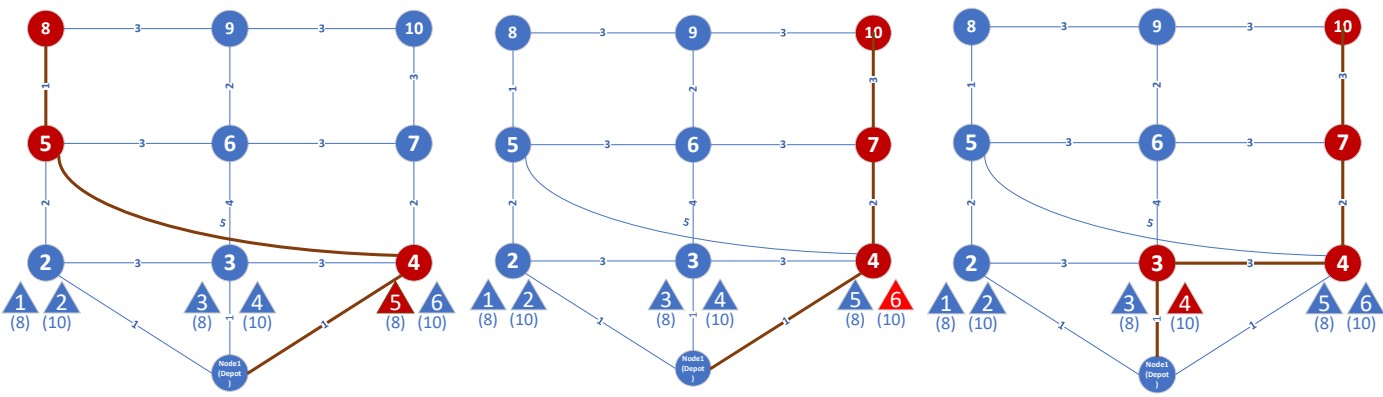

**Figure 8.** *Cont.*

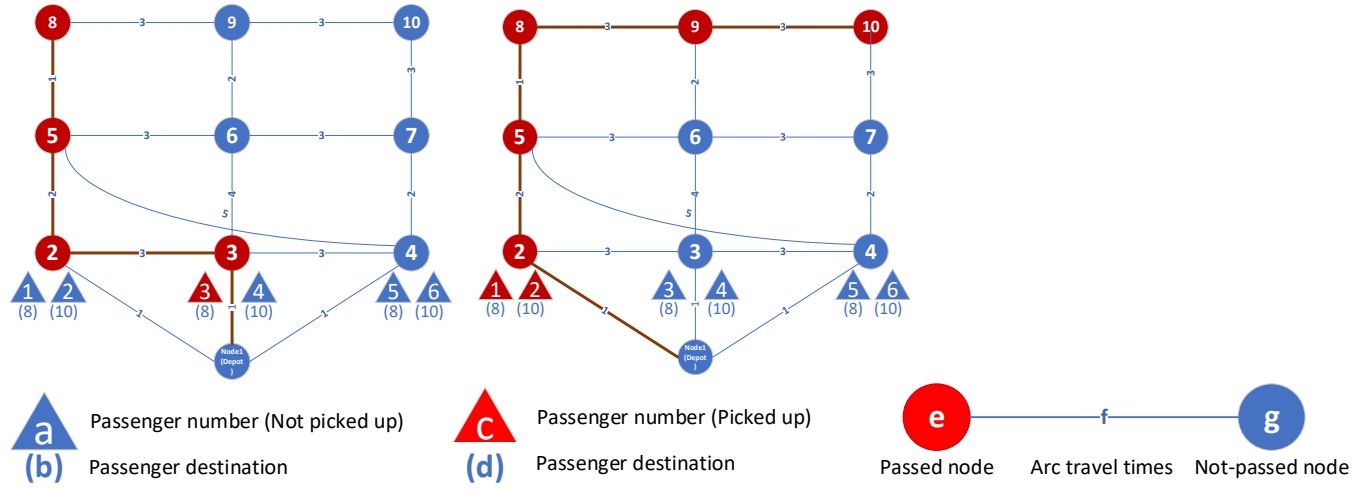

**Figure 8.** Solution 1 (Z1 = 5029, Z2 = 43) by AUGMECON and NSGA II.

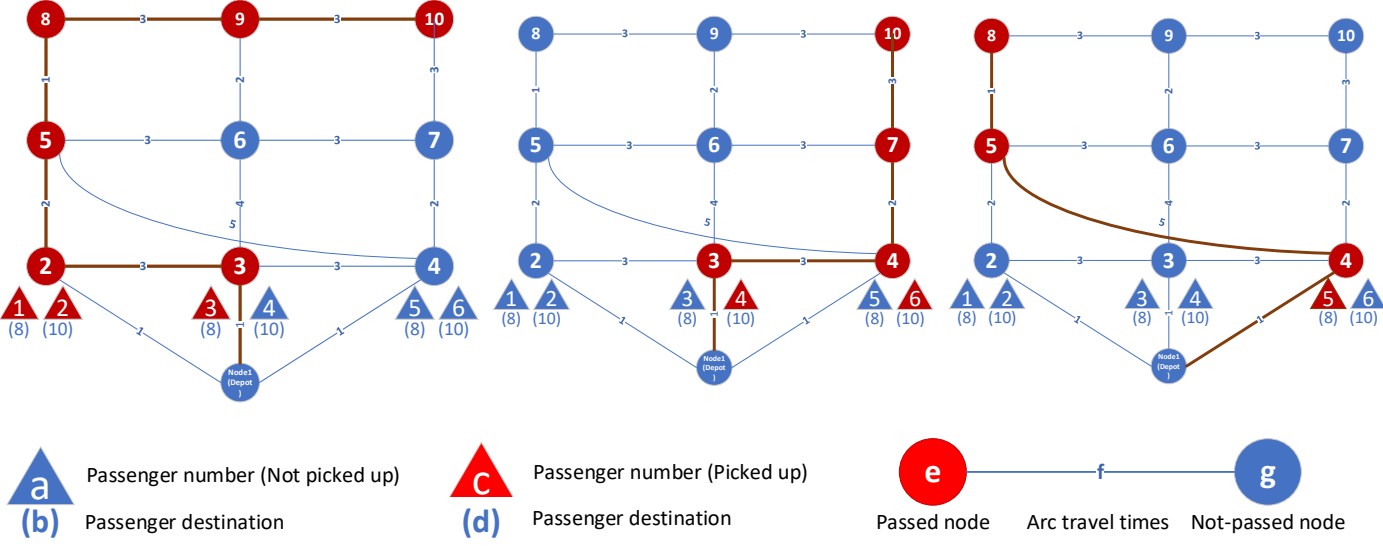

**Figure 9.** Solution 2 (Z1 = 3029, Z2 = 52) by AUGMECON and NSGA II.

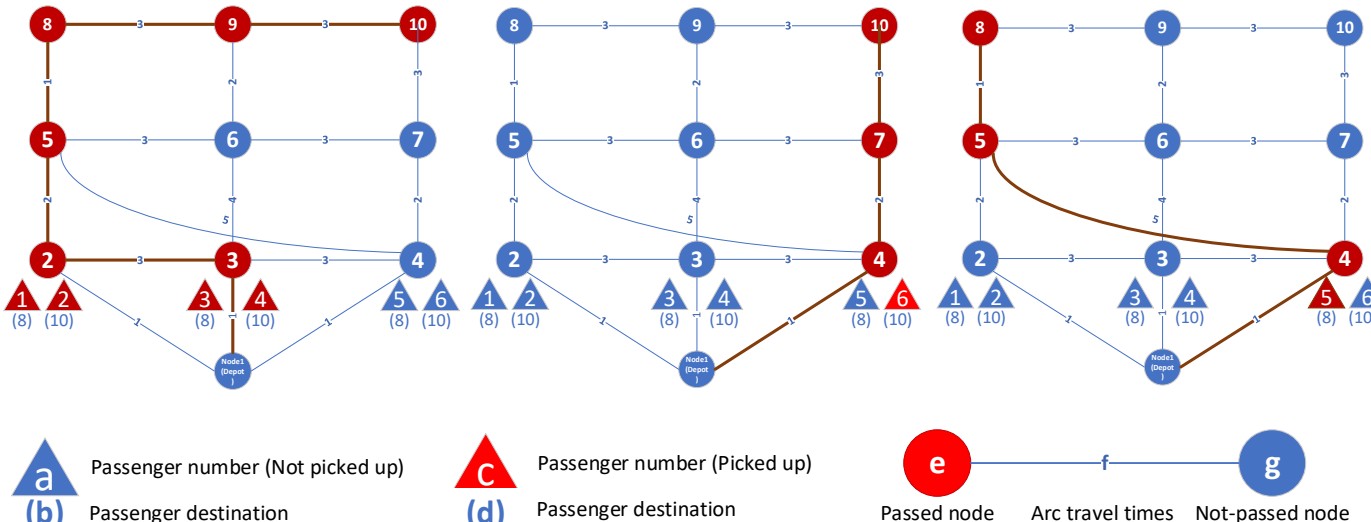

**Figure 10.** Solution 3 (Z1 = 3026, Z2 = 53) by NSGA II.

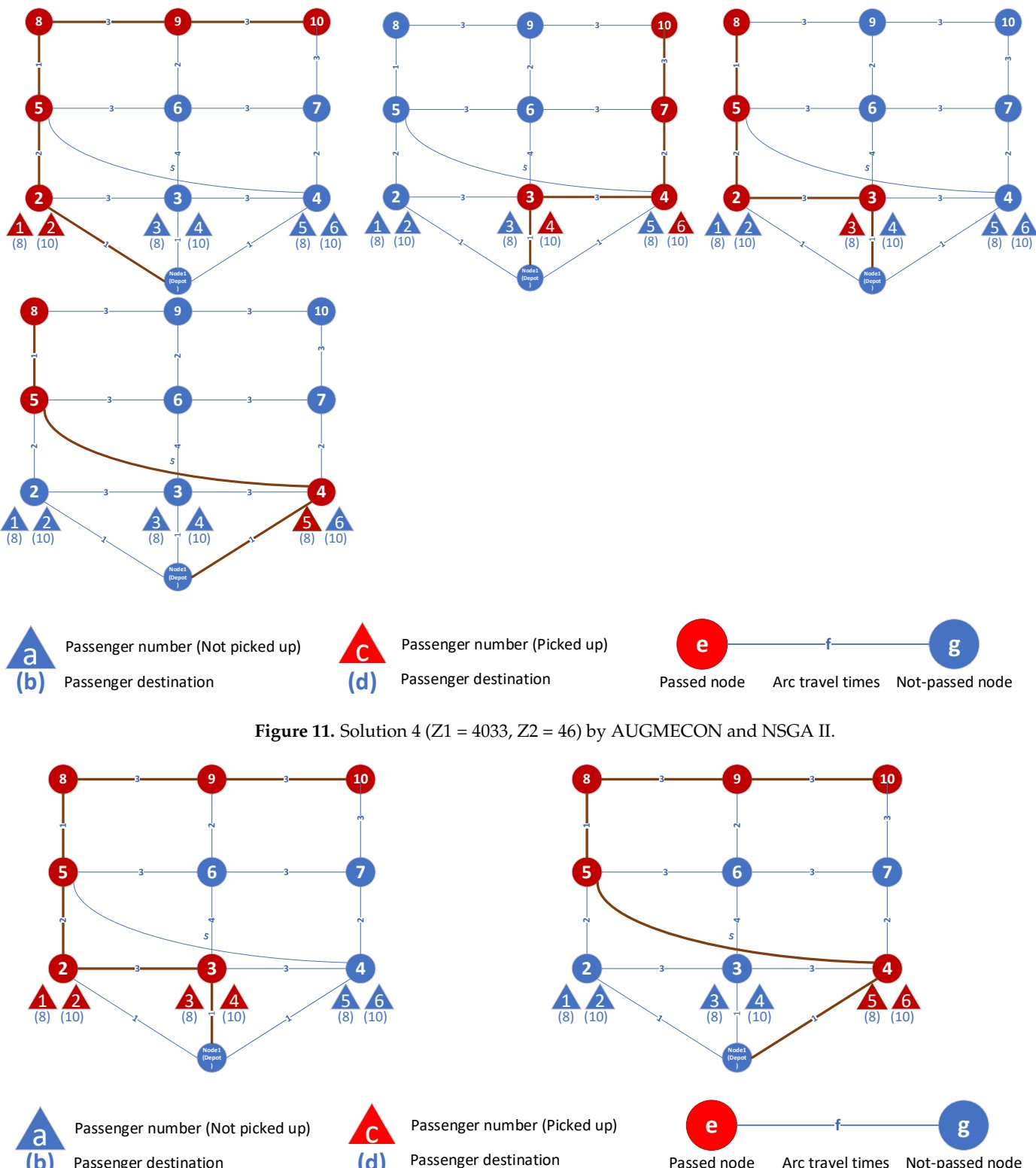

**Figure 11.** Solution 4 (Z1 = 4033, Z2 = 46) by AUGMECON and NSGA II.

**Figure 12.** Solution 5 (Z1 = 2026, Z2 = 60) by AUGMECON and NSGA II.

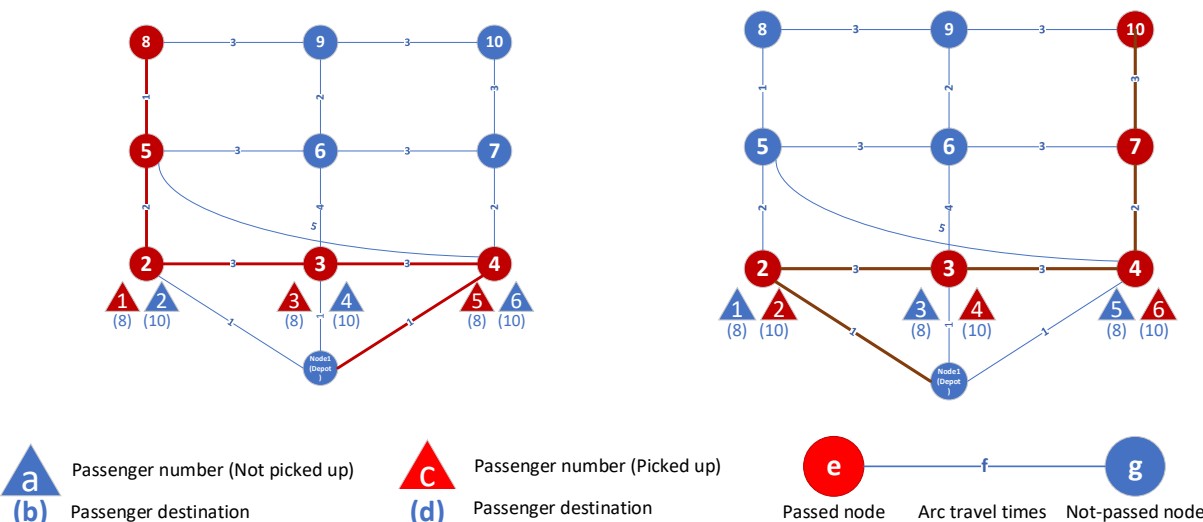

**Figure 13.** Solution 6 (Z1 = 2022, Z2 = 66) by AUGMECON and NSGA II.

b.    Case study

To demonstrate how NSGA II can be applicable in a real-world case, this paper uses data from the Sioux Falls network, a real-world transportation network available at the "TransportationNetworks" GitHub repository [49]. The data has 24 nodes and 76 links. The first node is considered a virtual depot, and nodes 1 to 20 are passenger nodes. Nodes 21 to 24 are supposed to be destination points. Figure 14 shows the map of Sioux Falls. A total of 439 passengers travels from nodes 1 to 20 (origin) to nodes 21 to 24 (destination).

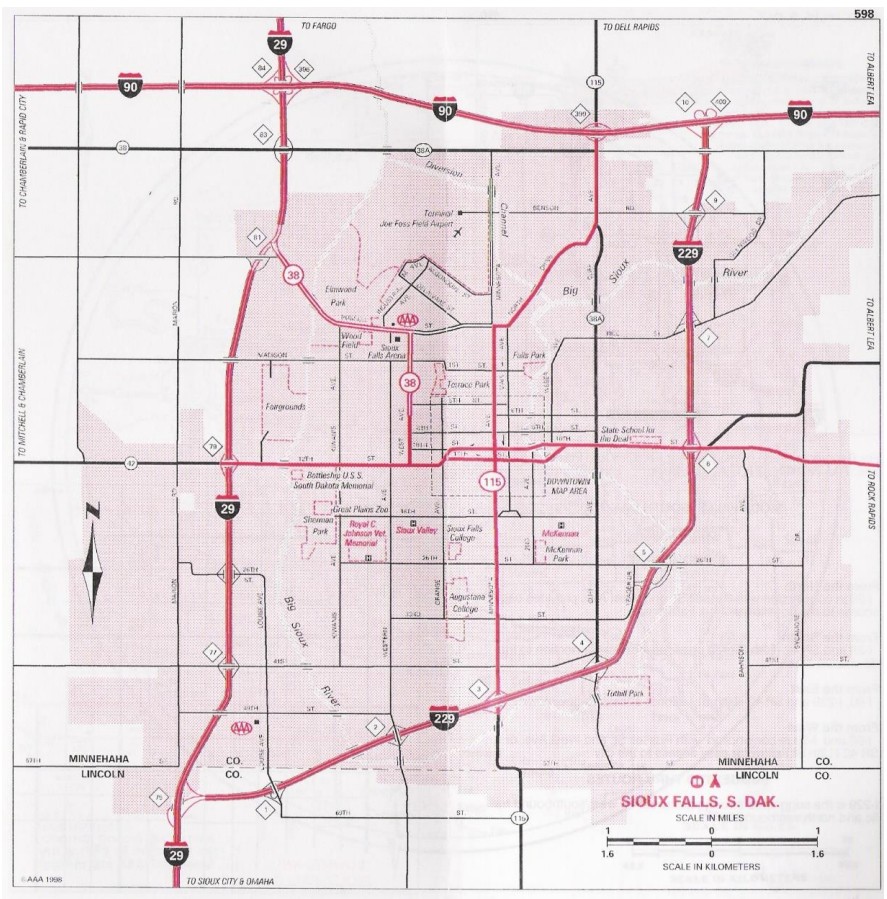

**Figure 14.** Sioux Falls network.

There are four different ways to evaluate the impact of ridesharing on the average travel time per person. The figure shows travel time per passenger (red line) and costs per passenger (blue column) over the different scenarios. In the first scenario, no ride-sharing strategy (vehicle capacity = 1) is applied. In these states, travel times per passenger are the lowest. In the non-ride-sharing scenario, there is only one solution since it is a single objective with a travel time objective function. The best optimal travel time in the non-ride-sharing scenario is 25.37 min, and the cost per passenger with fixed cost = 1000 is 1025.374. In the second through fourth scenarios, capacity and fixed cost are considered (2,1000), (3,1000), and (4,1000), respectively. The pareto fronts of these scenarios are shown in Figures 15–17. As expected, the first objective (total cost) has a high reverse correlation with second objective function (total passengers travel time).

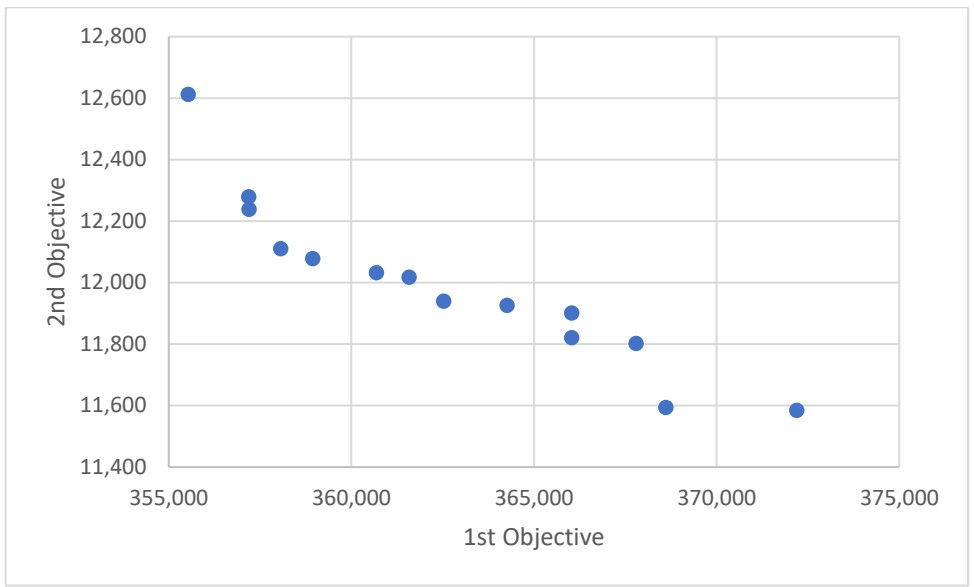

**Figure 15.** Pareto front for Capacity = 2 and Fixed Cost = 1000.

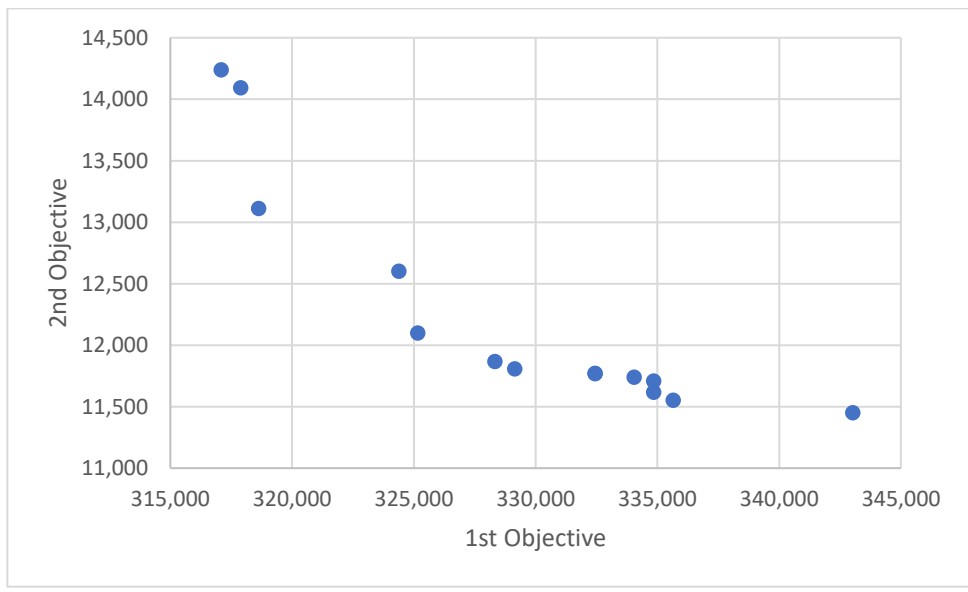

**Figure 16.** Pareto front for Capacity = 3 and Fixed Cost = 1000.

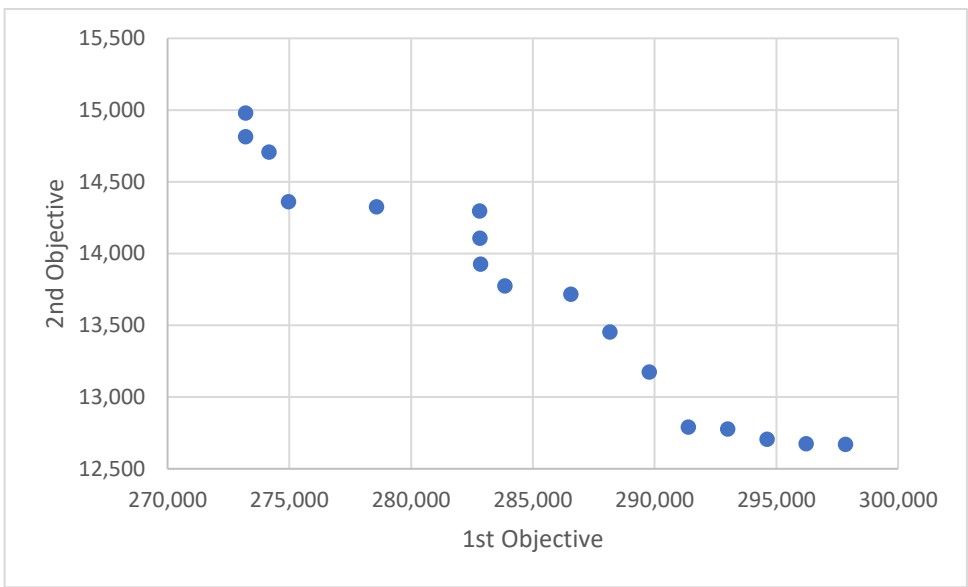

**Figure 17.** Pareto front for Capacity = 4 and Fixed Cost = 1000.

Figures 18–20 demonstrate the results of the pareto graph relationship between the cost and time from the passenger's perspective. In any scenario, a large variety of solutions are reachable in which the passengers can select the desirable combination of cost and time. These selections could be considered as a guideline for ridesharing service development to satisfy the passengers' expectations. When the capacity of the vehicles is increased, the total cost would be decreased while total reaching time would be increased. However, all solutions have a higher reaching time and lower cost than the condition without ridesharing Figure 21 shows the relationship between the capacity of the vehicles and the travel time per passenger. When the capacity of the vehicles is increased, more passengers are carried in a common vehicle, which increases the waiting time for passengers and consequently increases the travel time. Meanwhile, the increscent in reaching time is not generally ineligible and in the worst case, the reaching time exceeds less than %40 rather to no ridesharing strategy.

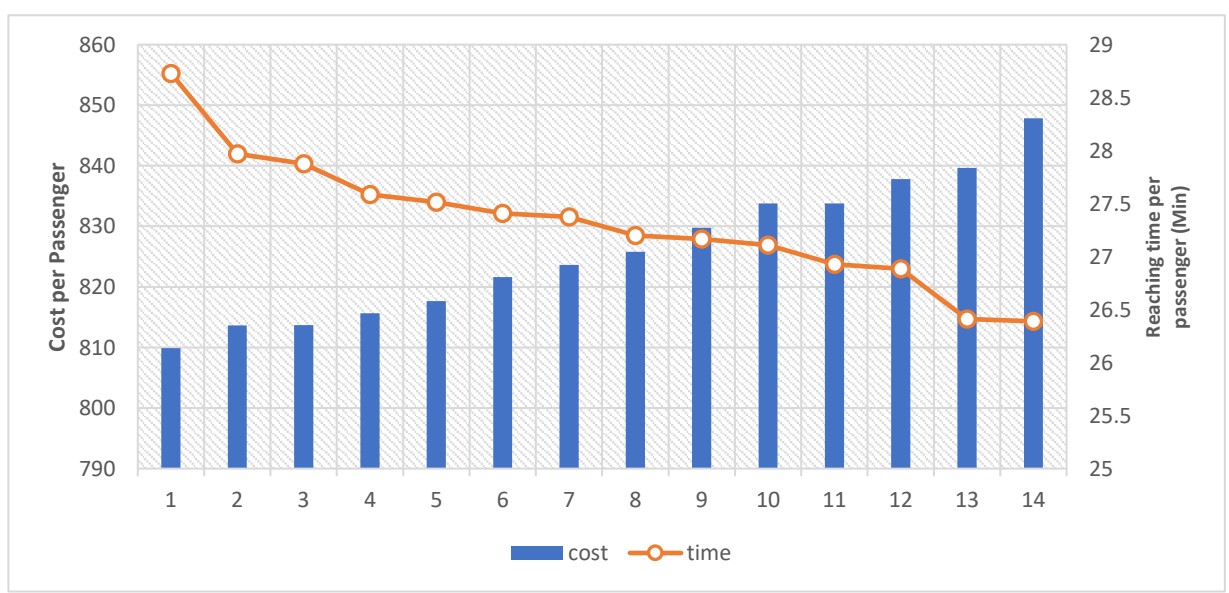

**Figure 18.** Pareto solutions (Capacity = 2, Fixed cost = 1000).

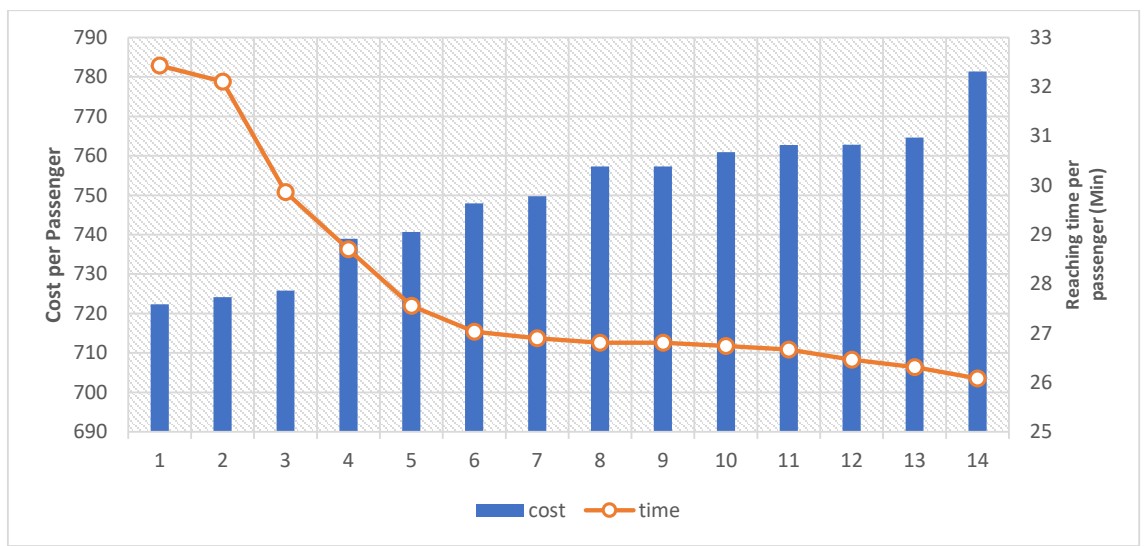

**Figure 19.** Pareto solutions (Capacity = 3, Fixed cost = 1000).

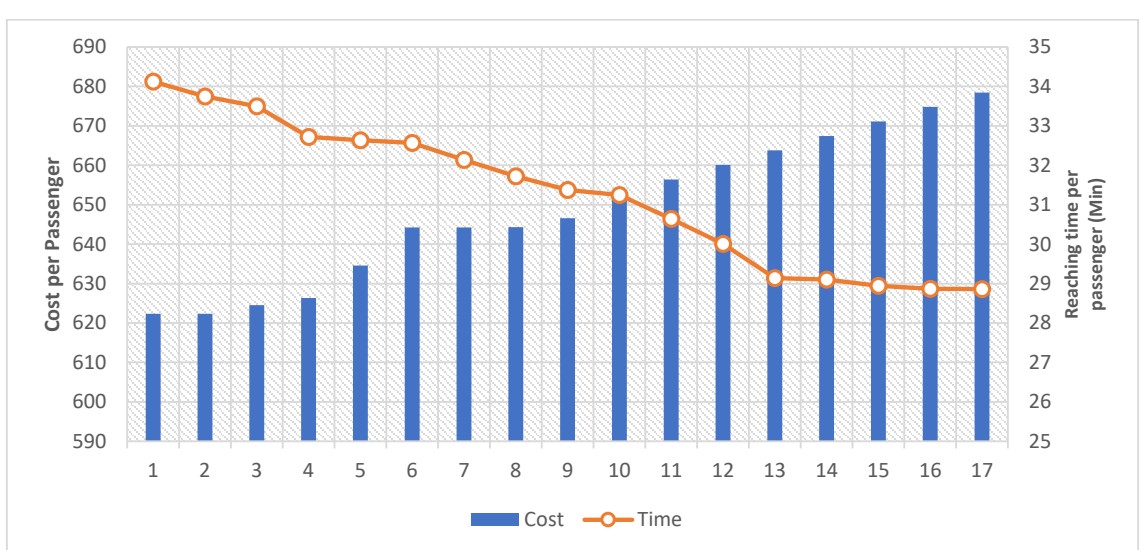

**Figure 20.** Pareto solutions (Capacity = 4, Fixed cost = 1000).

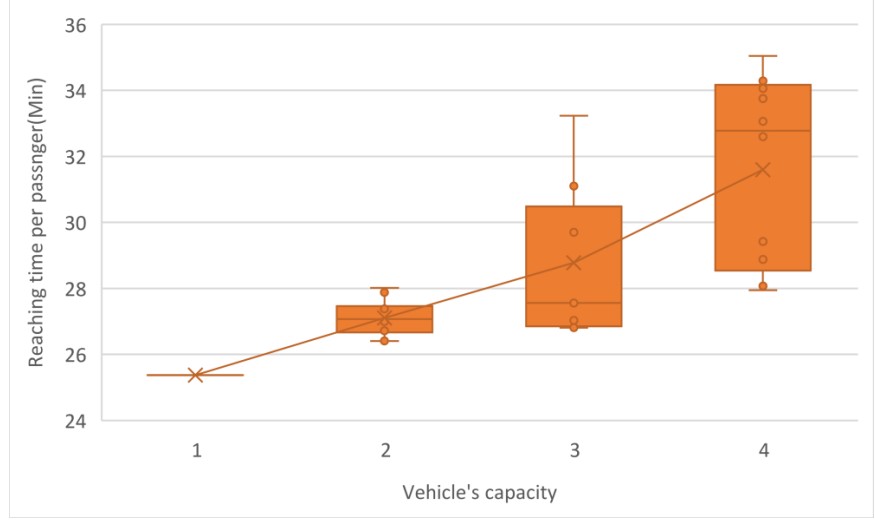

**Figure 21.** The relationship between reaching time and vehicle's capacity.

Compared to ridesharing, the cost of provisioning when no ridesharing occurs is highest. This is because each passenger was carried by a vehicle assigned exclusively to them, while if the vehicle capacity is increased, the number of the assigned vehicles would be less than the number of the passengers. Twelve scenarios are considered to compare the cost per passenger. In each scenario, two parameters are changed: capacity and fixed costs. Figure 22 shows the effects of the parameters on the cost per passenger.

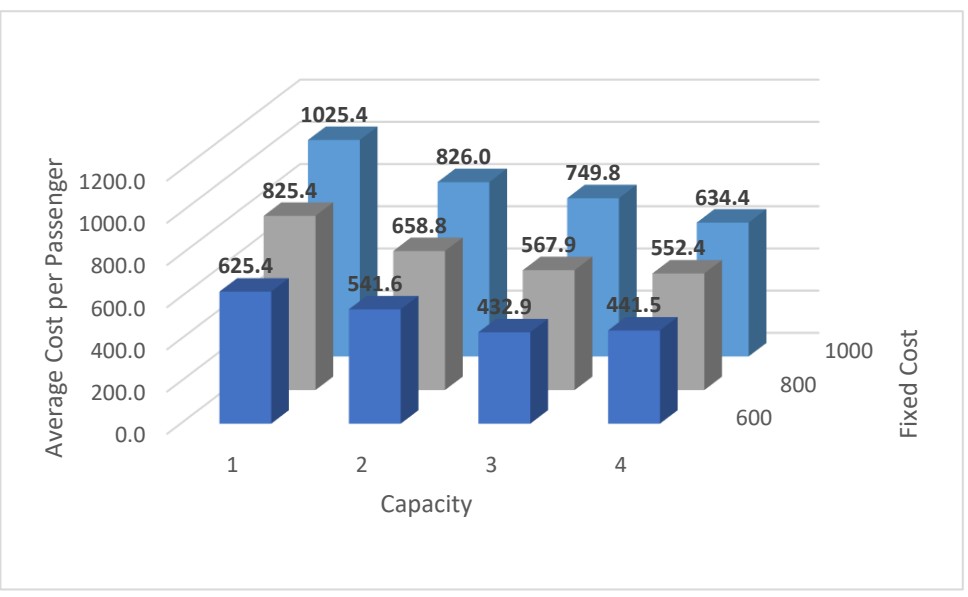

**Figure 22.** The relationship between average cost per passenger with capacity and fixed cost.

As you can see, when the number of passengers in the vehicle increases, the cost per passenger decreases. In current transportation systems, most vehicles have a capacity of four or more, but only one or two people use the car per trip so they pay the cost of vehicle with four capacities but the vehicle is utilized as a single or double seated car. By applying the ridesharing, the cost of trips would decrease about %40 in average. For example, the average cost per passenger for the vehicle with capacity = 3 and fixed cost = 800 is lower than for the vehicle with capacity = 1 and fixed cost = 600. Additionally, the average cost per passenger for a vehicle with an average capacity of four (4) passengers and a fixed cost of $1000 is almost the same as for a vehicle with an average capacity of one (1) passenger and a fixed cost of $600. In addition, increasing the average capacity would increase the travel time, so it is important for the provider to find a balance between average cost and travel time.

## 6. Conclusions

A bi-objective integer optimization model is developed that integrates vehicle assignment, vehicle routing, and passenger assignment in the context of a ride-sharing strategy that allows a vehicle to be used for more than one passenger to reduce travel costs. Two solution approaches are used to solve the model. The first approach is AUGMECON, an exact method, and the second approach is NSGA II, a heuristic algorithm. For small-scale problems, the results of the two methods are quite the same, while the runtime for NSGA II is much lower. The model can handle the demand of passengers with different origins and destinations with a common vehicle. In addition, passengers with different origins or destinations, or both, can be picked up by a common vehicle. This approach strikes a balance between passenger travel time and mobility provider costs. Mobility service provider costs include fixed costs related to the number of assigned vehicles and variable costs related to the distance traveled by the vehicles. Alongside passengers' benefits, ridesharing lead to reduction in traffic congestion and emission and could be a potential candidate for sustainable transportation development. A real-world case study from Sioux

Falls is cited to illustrate the applicability of the proposed model. Finally, various sensitivity analysis scenarios are conducted to determine the impact of vehicle capacity and fixed costs on passenger arrival time and ridership costs. In this case study, researchers found that ridesharing did not significantly affect travel time for passengers, but it did make it more economical to use the ride-sharing service. The proposed model could be beneficial for network design with ridesharing to find the optimum capacity of vehicles. It also supports mobility service providers to identify the fleet numbers to respond to passenger's expectations.

There are several possible future research areas for this work. One suggestion is to capture uncertainty by modeling stochastic situations such as passenger arrival times and travel times between depots. Additionally, the model could be extended by considering the depot's location as one of the outputs of the optimization model. Moreover, given that multiple depots are normally used in real-world train scheduling, this problem could be extended to handle multiple depots.

**Author Contributions:** Conceptualization, S.O.H.J.; Methodology, S.O.H.J.; Supervision, M.A.S.; Visualization, S.O.H.J.; Writing—original draft, S.O.H.J. All authors have read and agreed to the published version of the manuscript.

**Funding:** This research received no external funding.

**Institutional Review Board Statement:** Not applicable.

**Informed Consent Statement:** Not applicable.

**Data Availability Statement:** Some or all data, models, or code generated used during the study are available in a repository online in accordance with funder data retention policies (Sioux-Falls Transportation Networks. (n.d.). Retrieved from GitHub: https://github.com/bstabler/TransportationNetworks, accessed on 22 October 2021). Some or all data, models, or code that support the findings of this study are available from the corresponding author upon reasonable request (NSGA II MATLAB Code and AUGMECON GAMS code).

**Conflicts of Interest:** The authors declare no conflict of interest.

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
