# Peer review of "Bi Objective Peer-to-Peer Ridesharing Model for Balancing Passengers Time and Costs"

_sustainability, doi:10.3390/su14127443_

Round 1
Reviewer 1 Report
This study presents a model and two multi-objective methods for planning ride-sharing systems. Comments:
“The problem addressed in this paper is the NP-hard integer programming problem”: This is not a specific problem.
The background is written in a confusing manner, with time jumps without grouping studies in a different way as well. Overall, it is hard to follow, lacking a clear direction.
Moreover, the contents of Table 1 are not consistent. For instance, the “proposed method” column contains entries such as “ILP”, “combinatorial optimization” and “optimization problem”. Obviously, the latter are not solution methods. Furthermore, the contents of the column “investigated problem” are confusing as well. For instance, the term “flexible participation” is not very insightful. These terms should correspond to well-known problems or to problems clearly defined in the corresponding section.
The authors write: “to the best of our knowledge, no algorithm optimizes vehicle routing between different destinations.”. However, there are several recent papers that consider multiple destinations. For instance the following are relevant, among others:
Yao, R., & Bekhor, S. (2021). A Dynamic Tree Algorithm for Peer-to-Peer Ridesharing Matching. Networks and Spatial Economics, 21(4), 801-837.
Zhou, Z., & Roncoli, C. (2022). A scalable vehicle assignment and routing strategy for real-time on-demand ridesharing considering endogenous congestion. Transportation Research Part C: Emerging Technologies, 139, 103658.
Auad-Perez, R., & Van Hentenryck, P. (2022). Ridesharing and fleet sizing for On-Demand Multimodal Transit Systems. Transportation Research Part C: Emerging Technologies, 138, 103594.
Also, the authors cite Smet (2021), where multiple destinations are also considered. Therefore, the contribution of the paper becomes unclear. The authors should clearly address this and update the review section.
Notation is hard to follow. Variables should be defined in equations not text. Also, using 2 and 3 letters or an entire word for a variable is not suggested in general.
The selection of these algorithms is not thoroughly justified.
“Moreover, the numbers larger than the number of passengers are considered the division between the passengers assigned to each vehicle”: this sentence is hard to read and should be revised. Similarly, “Therefore, 5, 6, and 7 are considered as the division” is also not clear. Do the authors mean separators? Overall, the solution representation section is hard to read.
Table 4 is not shown fully as it does not fit in the page. It also contains several typos.
The different problems should first be introduced with the Pareto fronts presented after.
Figures 3 require legends to understand the symbols.
Figure 14 is hard to read. It would be better to use two axes to show the simultaneous effect of the two parameters.
In general, English should be significantly improved. A thorough proofread is required as well as rephrasing in several parts. For instance, the following phrases need correction (a very brief list of errors found):
to find a non-dominant solution (“non-dominated”)
none of the vehicles have to reverse themselves (this does not make sense)
MENG (Xintong, A critical overview of four-stage model under the background of the rise of ride-sharing, 2021) studied the characteristics (the tile of the paper appears in text)
Overall, the paper suffers from poor presentation and clarity, while lacks a clear contribution. There may be merit in the computational results, yet it is hard to appreciate given the presentation.
Reviewer 2 Report
In the reviewed paper the Authors presented an optimization model that integrates vehicle assignment, vehicle routing, and passenger assignment to find a non-dominant solution based on cost and time. The model allows a vehicle to be used multiple times by different passengers. The first objective seeks to minimize the total cost, including the fixed cost, defined as the supply cost per vehicle and the operating cost, which is a function of the distance traveled. The second objective is to minimize the time it takes passengers to reach their destination. This is measured by how long it takes each vehicle to reach the passenger’s point of origin and how long it takes to get to the destination. The proposed model is solved using the AUGMECON method and the NSGA II algorithm. The Authors presented a case study from Sioux Falls in order to validate the applicability of the proposed model. This study shows that ride-sharing helps passengers save money using mobility services without significantly changing their travel time. In my opinion, the paper can be published after taking into account the following remarks:
- the paper should be prepared according to the Sustainability journal paper template requirements,
- the paper topic is chaotic. Please write the title clearly and legibly,
- in the "Keyword" section, a keyword "sustainable transportation" should be added,
- in the Introduction section, the Authors described the ride-sharing services, also mentioning the bike-sharing system. Very good. However, the Authors did not mention that, from the point of view of transport services in cities, bike-sharing systems are often used where public transport stops no longer reach. It is a very important future of this system from the point of view of sustainable transport development. Authors should mention about this in the Introduction section while referring to recent scientific works in this field e.g. "External Environmental Analysis for Sustainable Bike-Sharing System Development, doi: 10.3390 / en15030791", "Approaching sustainable bike-sharing development: a systematic review of the influence of built environment features on bike-sharing ridership, /doi.org/10.3390/ su14105795 ". One short paragraph in the Introduction section will be enough,
- at the end of the "Introduction" section the Authors wrote what was the main aim of the paper as well as what was contained in each paper section. It is very good. But, the Authors also repeat the main aim of the paper at the end of the second section. It is redundant and should be deleted,
- the Authors used the acronyms without explanation, e.g. ILP. All used acronyms should be explained their meaning in the paper text in line with their first usage,
- section 3 is called "3. Mathematical Formulation". At the beginning of this section, we can find a sentence like follows: ..."This section presents a tailored mathematical formulation for our proposed problems." This sentence is redundant and should be deleted because the section name say readers what is inside this section,
- e.g. "???2" - it is the same when "j" is written as "subscript" in other equations?
- in the presented study, the Augmecon method is used as an exact method to obtain exact Pareto solutions. This method should be described in a more detailed way,
- what does it mean the colors used on the figures? It is difficult to describe on which figures because these figures doesn't have numbers,
- on the figure called "Figure 1- The Pareto front for a test problem" the legend explaining the used colors should be added,
- on the figure called "Figure 10-Pareto solutions (Capacity=2, Fixed cost=1000)" - there is a lack of name and unit of axis "x",
- there is a poor discussion about obtained results. It should be extended,
- is "6. Conclusion" should be "6. Conclusions".
Round 2
Reviewer 1 Report
The authors have revised the manuscript according to reviewer comments.